# Creating an atlas of the bone microenvironment during oral inflammatory-related bone disease using single-cell profiling

**Yi Fan[1], Ping Lyu[1], Ruiye Bi[2], Chen Cui[3], Ruoshi Xu[1], Clifford J Rosen[4], Quan Yuan[5]\*, Chenchen Zhou[6]\***

[1]State Key Laboratory of Oral Diseases, National Clinical Research Center for Oral Diseases, Department of Cariology and Endodontics, West China Hospital of Stomatology, Sichuan University, Chengdu, China; [2]State Key Laboratory of Oral Diseases, National Clinical Research Center for Oral Diseases, Department of Orthognathic and TMJ Surgery, West China Hospital of Stomatology, Sichuan University, Chengdu, China; [3]Hospital of Stomatology, Guanghua School of Stomatology, Sun Yat-Sen University, Guangdong Provincial Key Laboratory of Stomatology, Guangzhou, China; [4]Maine Medical Center Research Institute, Scarborough, United States; [5]State Key Laboratory of Oral Diseases, National Clinical Research Center for Oral Diseases, Department of Oral Implantology, West China Hospital of Stomatology, Sichuan University, Chengdu, China; [6]State Key Laboratory of Oral Diseases, National Clinical Research Center for Oral Diseases, Department of Orthodontics, West China Hospital of Stomatology, Sichuan University, Chengdu, China

**\*For correspondence:**
yuanquan@scu.edu.cn (QY);
chenchenzhou5510@scu.edu.
cn (CZ)

**Competing interest:** The authors declare that no competing interests exist.

**Abstract** Oral inflammatory diseases such as apical periodontitis are common bacterial infectious diseases that may affect the periapical alveolar bone tissues. A protective process occurs simultaneously with the inflammatory tissue destruction, in which mesenchymal stem cells (MSCs) play a primary role. However, a systematic and precise description of the cellular and molecular composition of the microenvironment of bone affected by inflammation is lacking. In this study, we created a single-cell atlas of cell populations that compose alveolar bone in healthy and inflammatory disease states. We investigated changes in expression frequency and patterns related to apical periodontitis, as well as the interactions between MSCs and immunocytes. Our results highlight an enhanced self-supporting network and osteogenic potential within MSCs during apical periodontitis-associated inflammation. MSCs not only differentiated toward osteoblast lineage cells but also expressed higher levels of osteogenic-related markers, including Sparc and Col1a1. This was confirmed by lineage tracing in transgenic mouse models and human samples from oral inflammatory-related alveolar bone lesions. In summary, the current study provides an in-depth description of the microenvironment of MSCs and immunocytes in both healthy and disease states. We also identified key apical periodontitis-associated MSC subclusters and their biomarkers, which could further our understanding of the protective process and the underlying mechanisms of oral inflammatory-related bone disease. Taken together, these results enhance our understanding of heterogeneity and cellular interactions of alveolar bone cells under pathogenic and inflammatory conditions. We provide these data as a tool for investigators not only to better appreciate the repertoire of progenitors that are stress responsive but importantly to help design new therapeutic targets to restore bone lesions caused by apical periodontitis and other inflammatory-related bone diseases.

## Editor's evaluation

Data from scRNA-Seq analysis demonstrated that acute inflammation stimulates periodontal stem cells to differentiate into osteoblast lineage cells to protect the alveolar bone. In murine models and patients with apical periodontitis, the genes and proteins associated with osteogenesis were enriched. The studies help us understand how MSCs respond to inflammation during apical periodontitis disease progression.

## Introduction

Oral diseases, particularly dental caries and periodontal diseases, affect 3.5 billion people worldwide (*Disease et al., 2018*). Untreated dental caries can directly lead to pulp necrosis and periapical lesions, resulting in apical periodontitis (AP). Individuals with at least one tooth affected by AP comprise up to 52% of cases, indicating that AP is a highly prevalent disease (*Tibúrcio-Machado et al., 2021*). Inflammation in the oral cavity can lead to destruction of surrounding periapical tissues and resorption of hard tissues, a consequence of the unbalanced interaction between infection and the immune response (*Gazivoda et al., 2009*; *Márton and Kiss, 2014*). Restoring and regenerating the destroyed periapical alveolar bone structures have always been a challenging task in clinical practice. Active inflammation, the tissue injury and the protective process all occur simultaneously in the setting of chronic AP (*Márton and Kiss, 2014*). Importantly, there is a complex assemblage of immune cell types involved in the pathogenesis, highlighting the importance of polymorphonuclear leukocytes, lymphocytes, and monocyte/macrophages in periapical defense (*Braz-Silva et al., 2019*; *Nair, 2004*). Notably, an increasing number of studies report the involvement of mesenchymal stem cells (MSCs) in the protective action that occurs during oral inflammatory diseases, whereby MSCs exert immunomodulatory effects and have regenerative potential (*Li et al., 2014*; *Márton and Kiss, 2000*; *Nair, 2004*). MSC markers such as CD44, CD73, CD90, CD106, and STRO-1 have been observed in human periapical inflammatory tissues (*Estrela et al., 2019*; *Liao et al., 2011*). Cells isolated from the inflamed periapical region were able to produce colony-forming unit-fibroblasts (CFU-Fs) with high-osteogenic capacity. It is also reported that interference with MSC mobilization toward the periapex region in an AP mouse model led to enlargement of lesions, accompanied by decreased wound healing markers and increased inflammatory cytokines (*Araujo-Pires et al., 2014*). These findings indicate the involvement of MSCs in the repair and regeneration of oral inflammatory-related bone lesions. They also suggest that MSCs present promising targets for treating bone lesions, with great potential for modulating inflammation and promoting tissue regeneration. However, most studies have surveyed whole tissues to understand the transcriptomic and cellular profile of these diseases. Specific cell populations and their regulatory molecules, as well as the interaction among different cell populations, remain far from clear. The advances in single-cell technologies offer an unbiased approach for identifying heterogeneous cell subsets, tracking the trajectories of distinct cell clusters and uncovering regulatory relationships between genes (*Hwang et al., 2018*; *Tang et al., 2009*). In this study, we collected mandibular alveolar bone samples from control and AP in mice and subjected them to single-cell RNA sequencing (scRNA-seq). The atlas of the mandibular alveolar bone explored the distinct cell subsets and their expression profiles relevant to AP. We also investigated the relationship between MSCs and immune cell subsets. The results reveal the role of a subset of MSCs in inflammation, which showed increased frequency and which formed a self-supporting network. Moreover, MSCs exhibited upregulated osteogenic potential, which was confirmed in transgenic mouse models and human patients with chronic AP. These results advance our understanding of heterogeneity and interactions of alveolar bone cells in the pathogenesis of inflammatory-related bone diseases. Defining key cellular subsets such as MSCs and their biomarkers in inflamed tissue will be important for identifying new therapeutic targets for oral inflammatory-related bone diseases.

## Results

### Single-cell transcriptional profiling identified 15 discrete populations in homeostasis and chronic AP samples

Individual cells were isolated from alveolar bone of healthy mice and mice with AP. We modeled AP using a well-established AP mouse model in which the mandibular first molar pulp was exposed and subsequently developed chronic AP over a 3 wk period (*Taira et al., 2019*). Bar-coded cDNA libraries from individual cells were obtained using the 10× Genomics Chromium Controller platform (*Zheng et al., 2017*; *Figure 1A*). The combined libraries from healthy and AP alveolar bone contained 15,148 individual cells. The median value of feature_RNA was between 1000 and 2000 (*Figure 1—figure supplement 1B*). After quality control filtering and removal of the batch effect between batches, the t-stochastic neighbor embedding (t-SNE) method was applied to reduce the dimensionality. Seurat's unbiased cluster detection algorithm identified 15 distinct cell populations (*Figure 1B and C*). Cluster-specific transcripts were utilized to annotate cell types with classic markers as described in a previous study (*Lin et al., 2021*). These included B cell (*Cd79a*), hematopoietic stem cell (HSC) (*Cd34*), MSC (*Col1a1*), natural killer (NK) cell (*Klrd1*), T cell (*Cd3g*), dendritic cell (*Siglech*), epithelial cell (*Epcam*), erythrocyte (*Hbb-bt*), macrophage (*Adgre1*), mast cell (*Fcer1a*), megakaryocyte (*Gp1bb*), monocyte (*Ly6c2*), myeloid progenitor (*Mpo*), neutrophil (*S100a8*), and pre-B cell (*Vpreb1*; *Figure 1E and F*). The top 20 enriched genes in each defined cluster were identified and compared (*Figure 1G*).

### AP led to significant changes in frequency and transcriptional expression of cell populations

All the identified cell clusters were present in both AP and control samples, but there were significant differences in the cellular compositions of particular clusters. T cell, B cell, NK cell, macrophage, epithelial cell, and MSC had significantly increased frequency in AP samples. Neutrophil, myeloid progenitor, monocyte, megakaryocyte, mast cell, HSC, and dendritic cell were markedly decreased (*Figures 2A and 1D*).

AP is a complex inflammatory process involving innate and adaptive immune responses (*Cotti et al., 2014*). A variety of inflammatory cells such as neutrophils, mast cells, monocytes, macrophages, and lymphocytes are involved in periapical lesions, highlighting the direct involvement of the immune response in the pathogenesis of AP (*Nair, 2004*). Neutrophils are important components in the acute phase of AP as a first line of defense. But they are also important in the progression of AP by interacting with microorganisms, leading to tissue damage and chemotaxis (*Braz-Silva et al., 2019*). Single-cell differential expression analysis revealed that the most significantly enriched genes in neutrophils were various proinflammatory chemokines and cytokines. These included C-X-C motif chemokine ligand 2 (*Cxcl2*), C-C motif chemokine ligand 6 (*Ccl6*), NLR family pyrin domain containing 3 (*Nlrp3*), and Inter-leukin-1β (*Il1b*). Notably, we found that C-C motif chemokine receptor like 2 (*Ccrl2*) was upregulated in neutrophils during AP (*Figure 2D*). It is responsible for the innate defense against pathogens and is also involved in the regulation of neutrophil migration (*Del Prete et al., 2017*; *Kolaczkowska and Kubes, 2013*; *Mantovani et al., 2011*).

Mast cells, monocytes, and macrophages have critical roles in the inflammatory infiltrate during chronic AP (*Braz-Silva et al., 2019*). The production of Interleukin-6 (Il6) was present in these cell populations with the highest expression level in mast cells. The pro-inflammatory cytokine IL-1β is a key mediator of host response to microbial infection and is associated with the persistence of AP (*Morsani et al., 2011*; *Ng et al., 2008*). We found *Il1b* transcripts in a series of cell types, such as monocyte, macrophage, mast cell, and neutrophil. Of these, macrophages had the highest Il1b expression. Another major cytokine, tumor necrosis factor (*Tnf*; *Cotti et al., 2014*), was detected in immunoresponsive cell clusters, such as monocyte, macrophage, mast cell, myeloid progenitor, neutrophil, and HSC, with the highest expression observed in the monocyte population (*Figure 2B*).

Furthermore, gene signatures from monocytes showed that the interferon-induced transmembranes (IFITMs) protein 1 and 2 (*Ifitm1* and *Ifitm2*; *Figure 2E*) were upregulated the most during AP. These factors have been associated with signal transduction of anti-inflammation activity in the immune system (*Yánez et al., 2020*). We also detected upregulated expression levels of *Ccl9* in the monocyte population from AP. Ccl9 is an important cytokine and is involved in the survival of osteoclasts during the destruction of the periapical bone (*Silva et al., 2007*). Also, genes coding for pro-inflammatory

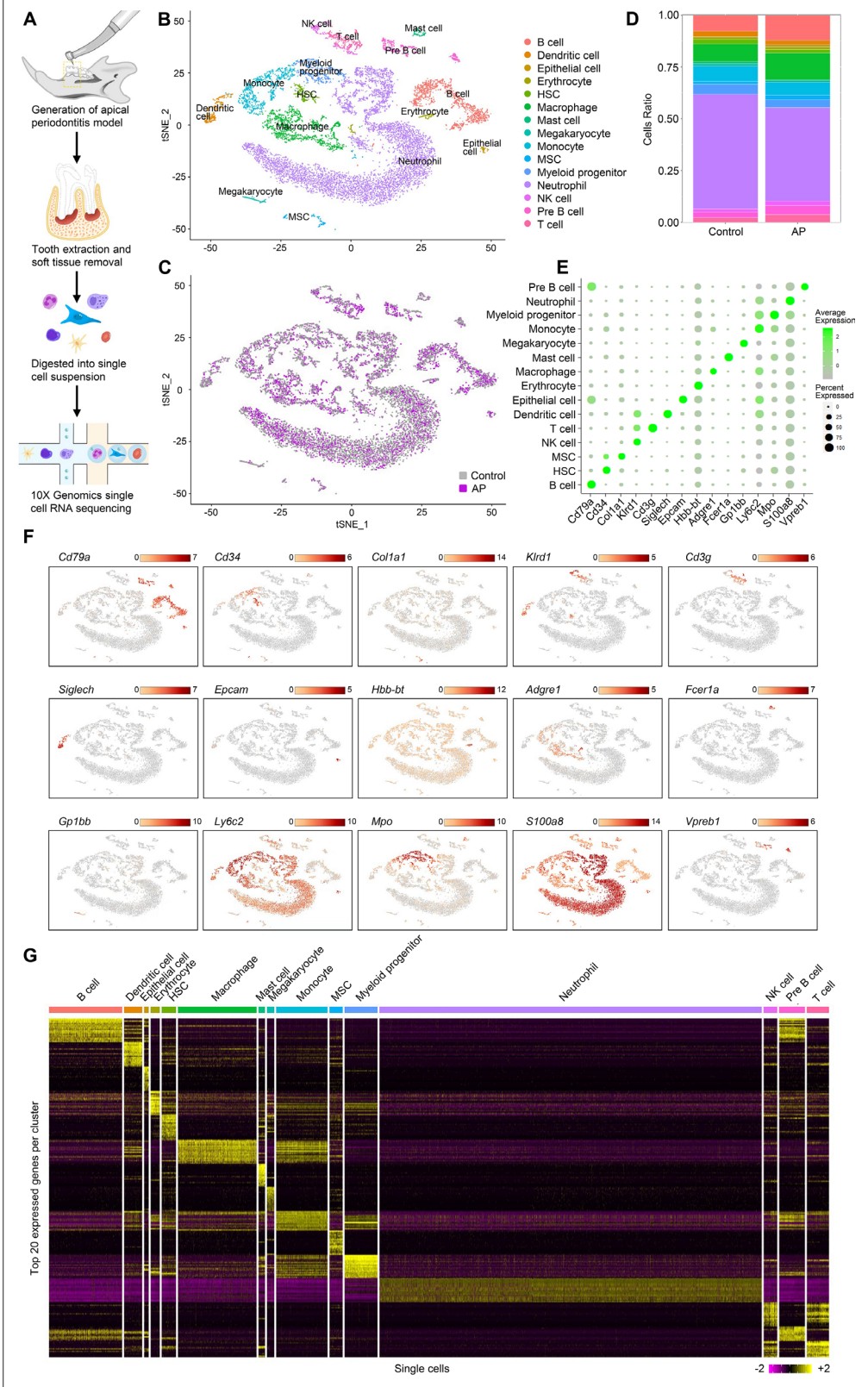

**Figure 1.** Identification of the single-cell atlas of alveolar bone using scRNA-sequencing (scRNA-seq) and unbiased clustering. (**A**) Schematic diagram of the experimental design. (**B–C**) t-Stochastic neighbor embedding (t-SNE) representation of aligned gene expression data in single cells extracted from mandibles of control mice (n=8340) and apical periodontitis (AP) mice (n=6808) showing 15 distinct clusters and cellular origin. (**D**) Relative

*Figure 1 continued on next page*

*Figure 1 continued*

abundance of 15 cell populations composing alveolar bone under healthy and AP conditions. (**E**) Expression of gene markers in distinct cell types. (**F**) Gene expression patterns projected onto t-SNE plots of marker genes. Scale: log-transformed gene expression. (**G**) Heatmap showing the 20 most upregulated genes (ordered by decreasing Padj value) in each cluster defined in B. Scale: log2 fold change.

The online version of this article includes the following figure supplement(s) for figure 1:

**Figure supplement 1.** Quality control results of the sequencing data.

calcium-binding S100 family proteins such as *S100a9* and *S100a11* had increased expression in the AP monocyte cluster (***Figure 2E***). Previous reports demonstrated that macrophages are capable of secreting pro- and anti-inflammatory substances which act on the development and repair of the AP lesions (***Italiani and Boraschi, 2014***; ***Shapouri-Moghaddam et al., 2018***). Indeed, several genes encoding pro-inflammatory mediators, including *Cxcl2*, *Cxcl16*, *Il1a*, and *Ptgs2*, were upregulated in macrophages from AP samples (***Figure 2C***). Expression of anti-inflammatory-associated genes such as *Ifitm1* and *Ifitm2* was significantly increased in the AP macrophage cluster compared to control cells. Furthermore, Fcγ receptor IIB (*Fcgr2b*) was markedly upregulated in macrophages from AP samples. Fcgr2b is expressed in most tissue-resident macrophages (***Gautier et al., 2012***) and functions to inhibit Fcγ-dependent phagocytosis. It also inhibits release of cytokines such as IL-6, TNF-α, IL-1α, as well as neutrophil chemotactants (***Clatworthy and Smith, 2004***; ***Espéli et al., 2016***). In addition, expression of Apolipoprotein E (*Apoe*), which can suppress the pro-inflammatory response (***Jofre-Monseny et al., 2007***), was significantly increased in the macrophage population (***Figure 2C***). These data indicated the activation of anti-inflammatory factors by macrophages during local inflammation by AP.

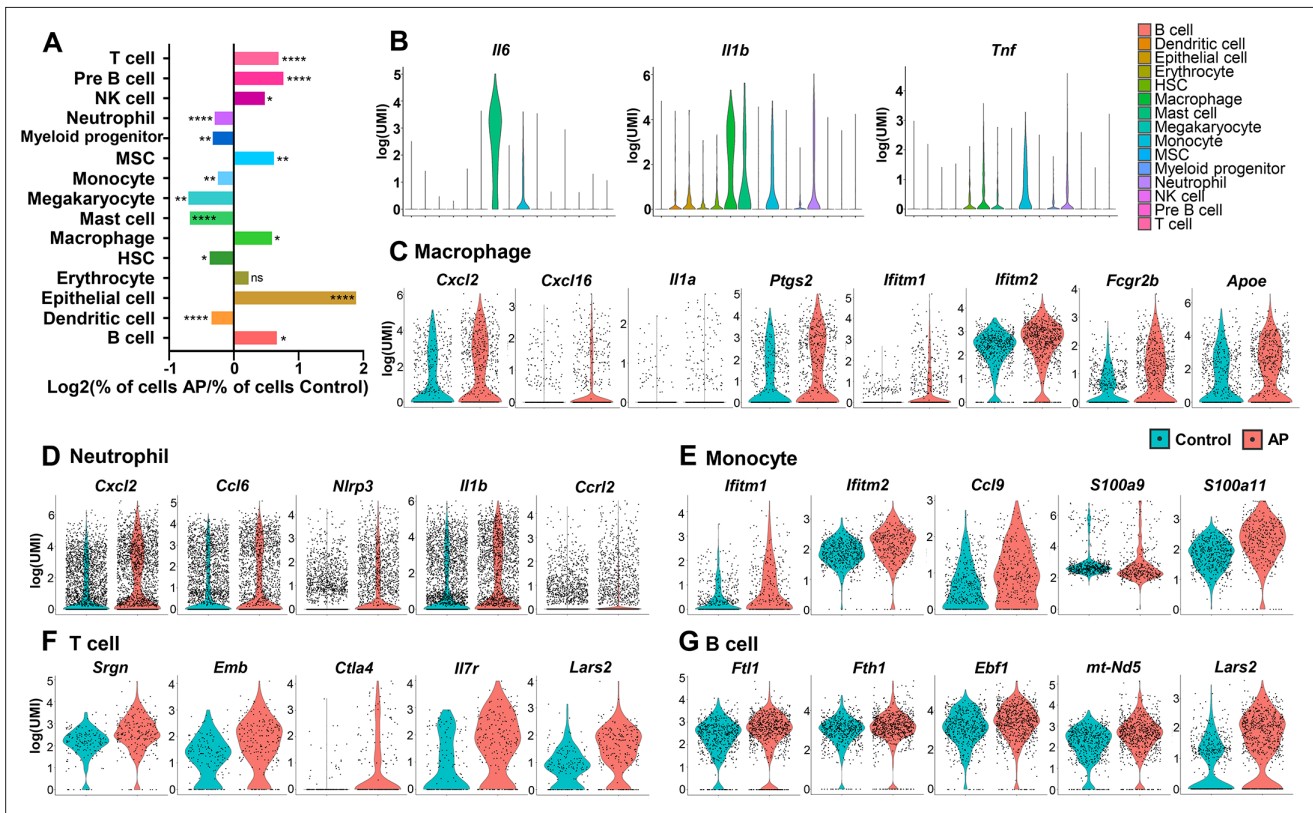

**Figure 2.** Changes in frequency and transcriptional expression pattern in each cell population from control and apical periodontitis (AP) groups. (**A**) Bar plot of cells per cluster (AP versus control). Normalization to overall number of inputs per condition. Fisher's exact test with Bonferroni correction was used. *p<0.05, **p<0.01, and ****p<0.0001. All data were exhibited as mean ± SEM. (**B**) Violin plots of cluster-specific expression of representative genes. (**C–G**) Violin plots showing genes that significantly changed in each cluster from control and AP.

The major classes of lymphocytes are T lymphocytes, B lymphocytes, and the NK cells. T and B lymphocytes comprise the majority of the inflammatory infiltrate in AP (*Graunaite et al., 2012*). A significant increase in the expression of inflammatory-associated genes, such as *Srgn*, *Emb*, *Ctla4*, and *Il7r* could be observed in the AP T cell population (*Figure 2F*). In the AP B lymphocytes cluster, inflammation-responsive genes (*Fth1*, *Ftl1*, *Ebf1*, and *mt-Nd5*) were upregulated (*Figure 2G*). Interestingly, *Lars2*, the gene encoding a mitochondrial leucyl tRNA synthase (*Carminho-Rodrigues et al., 2020*; *'t Hart et al., 2005*), was significantly upregulated in T and B lymphocytes, indicating changes in mitochondrial metabolism in both clusters.

## Inflammation induces osteoclasts differentiation leading to periapical alveolar bone destruction

Bone destruction is a major pathological factor in chronic inflammatory diseases such as AP. Various cytokines including TNF-α, IL-1α, and IL-6 were released by immunocytes to recruit the osteoclast precursors and induce the maturation of osteoclasts (*Lyu et al., 2022*). We have detected osteoclast markers including *Ctsk*, *Acp5*, *Mmp9*, and *Nfatc1* by scRNA-seq. Moreover, *Csfr1*, *Cx3cr1*, *Itgam*, and *Tnfrs11a* were used to identify osteoclast precursors. Markers of osteoclast and osteoclast precursors were highly expressed in the clusters of monocyte and macrophage (*Figure 3A and B*). Gene Ontology (GO) analysis showed that inflammation related immune reactions and bone resorption activity were significantly enriched in macrophage cluster (*Figure 3C*). To further study the differential trajectory of osteoclasts, pseudotime analysis was performed for the clusters of macrophage and monocyte. Two independent branch points were determined, and five monocyte/macrophage subclusters were scattered at different branches in the developmental tree (*Figure 3D and G*). The results showed that the monocyte cluster differentiated into the macrophage cluster (*Figure 3E*). During this trajectory, the gene expression pattern across pseudotime showed that osteoclastic genes, such as *Ctsk*, *Acp5*, *Mmp9*, *Atp6v0d2*, and *Dcstamp*, were progressively elevated (*Figure 3F*). Of note, we have observed a branch which was highly positive for *Ctsk* and *Acp5* (*Figure 3H*), indicating the mature osteoclasts were differentiated from monocyte/macrophage lineage and contributed to inflammatory bone resorption during AP. We have also analyzed the expression of osteoclast related genes using the bulk RNA-seq library built on mandibular samples extracted from mice with AP. Markers of osteoclast and osteoclast precursors were significantly upregulated, confirming the osteoclasts activity in the inflammatory-related bone lesion (*Figure 3I*).

## AP leads to reduced transcriptionally inferred cellular interactions with an increased self-supporting network in MSCs

We next sought to characterize the cell-cell communication related to the perturbation of signaling pathways detected in the AP samples by employing CellphoneDB (*Efremova et al., 2020*; *Nagai et al., 2021*). We identified a close interaction among MSC, macrophage, and dendritic cells under homeostasis conditions (*Figure 4A*). A similarly close communication was found among MSC, macrophage, and dendritic cells under inflammatory conditions (*Figure 4B*). Next, we compared the differential cell-cell interaction (CCI) network between AP and control samples using CrossTalkeR (*Nagai et al., 2021*). The results suggested that AP is associated with an overall decrease in cellular interactions. However, MSC intercellular communication with mast cells and monocytes is upregulated and accompanied by the highest number of interactions within MSCs themselves (*Figure 4C*). These results indicate that, although cell populations lose their normal physiological interactions, MSCs were able to establish a self-interacting network and coordinate with certain types of cells during chronic AP. Next, we ranked the individual ligands by the number of their interactions. Inflammatory-related proteins (*Lgals9*, *Tnf*, and *Ccl4*), extracellular matrix protein (*Fn1*), and protein involved in biomineralization (*Spp1*) were among the highest interactions. Also, *Tgfb1*, *Vegfb*, and *Vegfa* were in the top 10 most abundant ligands (*Figure 4D*). Bar plots were also generated to display the top 10 upregulated gene/cell pairs, showing that inflammation associated genes (*Tnf*/neutrophil, *Ccl3*/mast cell, *Ccl3*/monocyte, and *Il1b*/macrophage) and matrix related genes (*Sele*/MSC, *Fn1*/MSC, and *Fn1*/monocyte) were the most influential ligands during AP when compared to control state (*Figure 4E*). We used a Sankey plot to further focus on MSC-related interactions (*Figure 4F*). The results indicated that Sele was primarily directed by MSC toward the MSC cluster via multiple receptors including Glg1, Selplg, and Cd44. Moreover, Fn1 was secreted by MSC, monocyte, macrophage, and mast cells toward MSCs

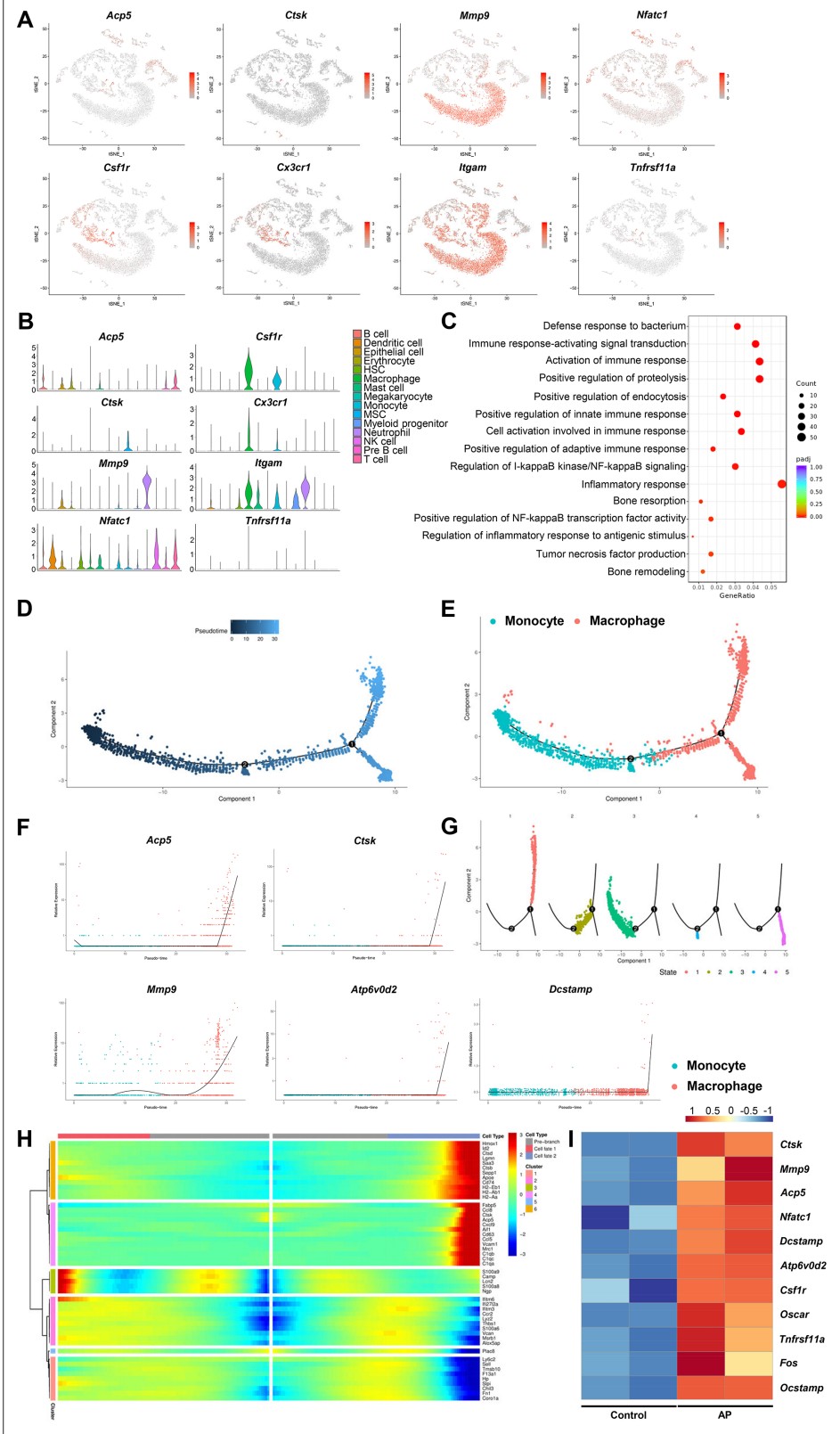

**Figure 3.** Inflammatory-related bone resorption under apical periodontitis (AP) situation. (**A**) The expression levels of markers of osteoclasts and osteoclast precursors. (**B**) Violin plots of the expression of osteoclastogenesis genes. (**C**) Gene Ontology (GO) enrichment analysis of the biological functions of macrophage cluster. (**D**) Trajectory order of the monocyte/macrophage populations by pseudotime value. (**E**) The differentiation trajectory of monocyte and

*Figure 3 continued on next page*

*Figure 3 continued*

macrophage clusters presented on a t-stochastic neighbor embedding (t-SNE) visualization. (**F**) The expression patterns of osteoclast markers during the trajectory of monocyte/macrophage populations. (**G**) Distribution of monocytes/macrophages on the developmental tree by clusters. (**H**) Heatmap of differential genes of three states. (**I**) Heatmap of genes associated with osteoclastogenesis in bulk RNA-seq analysis.

The online version of this article includes the following figure supplement(s) for figure 3:

**Figure supplement 1.** The expression pattern of markers of M1-like macrophages and M2-like macrophages in macrophage cluster.

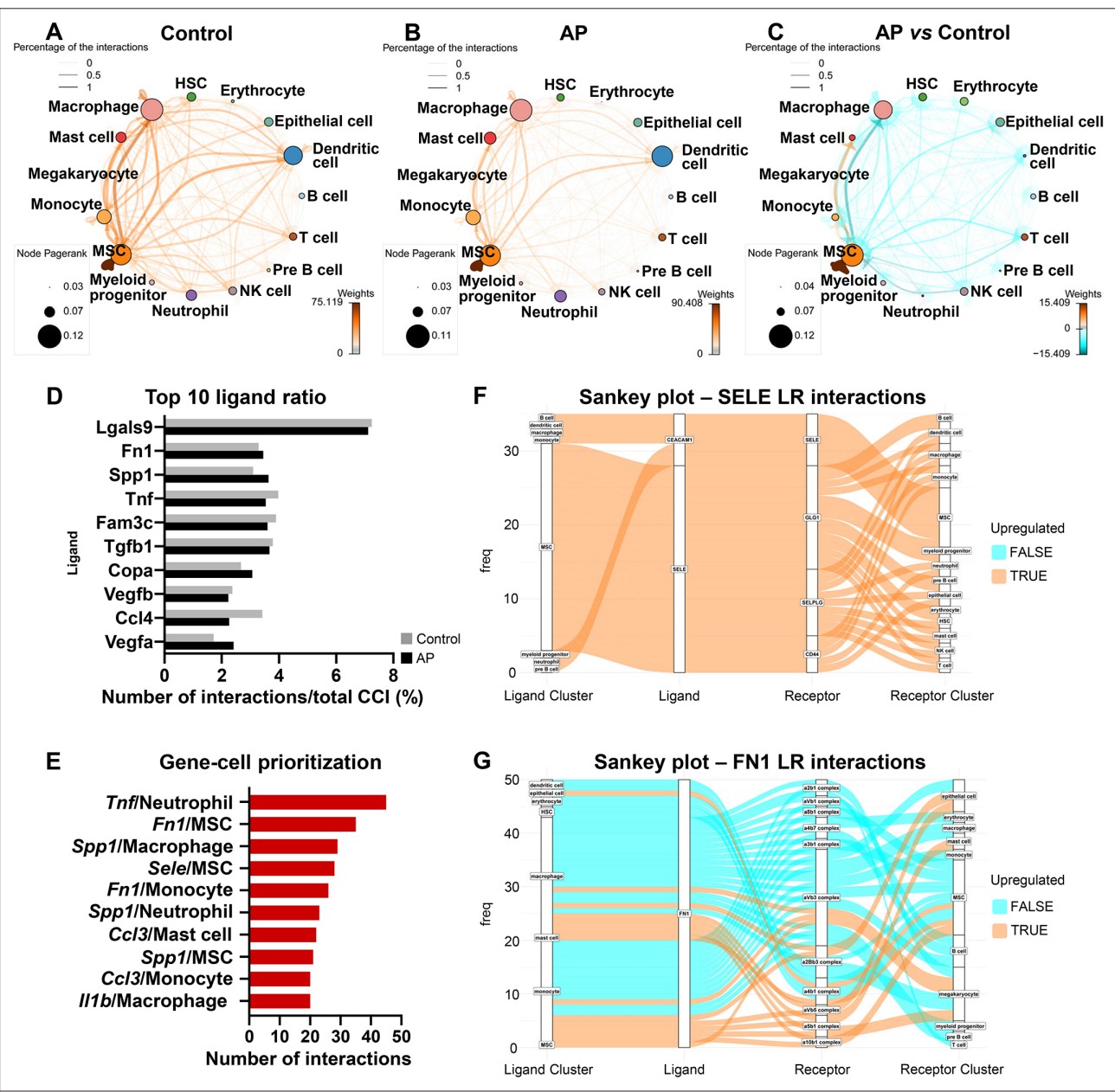

**Figure 4.** Apical periodontitis (AP) suppressed transcriptionally inferred cellular interactions and increased a self-supporting network within the mesenchymal stem cell (MSC) cluster. (**A–C**) Network plot of ligand-receptor activity in control (**A**), AP (**B**), and AP versus control (**C**). (**D**) Bar plot of top 10 most abundant ligands in all inferred ligand-receptor interactions. (**E**) Ranking of ligand/source regarding communication gains in AP state. (**F and G**) Sankey plot listing all predicted source, receptor, and receiver interactions associated with Fn1 and Sele.

(*Figure 4G*). Of note, Sele and Fn1 are important cell adhesion molecules that mediate cell homing and migration (*Frenette et al., 1998*; *To and Midwood, 2011*). This was in accordance with the increased frequency of the MSC cluster (*Figure 2A*) as well as the upregulated cellular interaction among MSCs and other cell populations.

## scRNA-seq based identification of AP-associated MSC population

MSCs represented the non-immune cell population in alveolar bone, constituting 1.76% of total identified cells. This cell population could be decomposed into four subclusters (*Figure 5A*). The most dominant subcluster was characterized by high expression of *Prrx1*, platelet-derived growth factor receptor β (*Pdgfrb*), and hematopoiesis supporting factors such as C-X-C motif chemokine 12 (*Cxcl12*) and angiopoietin (*Angpt1*). It was also characterized by osteogenic-related markers, such as Runt-related transcription factor 2 (*Runx2*), *Sp7*, and was thus classified as the MSC_osteolineage cells (OLCs). The other subclusters were identified as MSC_endothelial (*Cdh5*+), MSC_neurological (*Plp1*+), and MSC_inflammatory (*S100a8/S100a9*+; *Figure 5A and B*). The cell composition of the MSC cluster indicated an expansion of MSC_inflammatory subcluster during AP, whereas the MSC_neurological subcluster was reduced (*Figure 5C*). Of note, we performed lineage tracing experiments using *Prrx1^Cre^*, *Sp7^Cre^*, *Lepr^Cre^*, and *Gli1^CreER^* transgenic mouse models to identify stem cells in alveolar bone and the surrounding periodontal tissues (*Cui et al., 2020*; *Fan et al., 2022*; *Men et al., 2020*; *Zhang et al., 2020*). The results suggested increased numbers of Prrx1+ and Sp7+ MSC_OLCs as well as upregulated Gli1+ and Lepr+ periodontal ligament stem cells (PDLSCs) in AP lesions (*Figure 5—figure supplement 1A*). We compared the MSC marker genes, identified above, among the four subpopulations. Their respective GO enrichment analysis confirmed that there are four specific MSC populations. MSC_OLCs were enriched for ossification, connective tissue development, skeletal system development, and osteoblast differentiation. MSC_endothelial cells displayed enrichment for angiogenesis, blood vessel morphogenesis, and endothelium development. MSC_neurological cells were enriched for myelination, ensheathment of neurons, and axon ensheathment. GO term analyses suggested highly specialized functional features of MSC_inflammatory cells, including regulation of protein export from nucleus, electron transport chain, mitochondrial ATP synthesis coupled electron transport, and oxidative phosphorylation (*Figure 5—figure supplement 2A*).

## AP stimulates MSC differentiation toward osteoblast lineage cells

We next compared the transcriptome data from the MSC cluster between AP and control groups. Among the top upregulated genes, we observed a significant difference in the single cell expression levels of OLC-associated transcripts, such as secreted protein that is acidic and rich in cysteine (*Sparc*), *Col1a1*, *Col1a2*, *Bglap*, *Bglap2*, and *Postn*, accompanied by a trend toward increased *Runx2* expression (*Figure 5D*). We verified their expressions by real-time quantitative PCR and bulk RNA sequence analysis. The results revealed that *Sparc*, *Col1a1*, *Col1a2*, and *Runx2* were significantly upregulated, and *Bglap* tended to increase during AP (*Figure 5—figure supplement 3A, B*). In vivo experiments confirmed the upregulated expression of osteogenic-related markers, such as Sparc and Col1a1 in the AP group. There were increased numbers of Sparc+/Sp7+ or Col1+/Sp7+ cells embedded in the bone matrix, implying the protective function of MSC to differentiate into osteoprogenitors and osteoblasts under AP conditions (*Figure 5E and F*). Lineage tracing analysis further revealed that Gli1+ PDLSCs migrated to AP lesions and differentiated into Runx2+-osteoblasts (*Figure 5G*). Moreover, using *Lepr^Cre^;Rosa26^Ai14^;Col2.3-GFP* mouse model, we identified more Lepr+/Col2.3+ cells in the alveolar bone, confirming that inflammation could stimulate the progenitor cells in the adjacent bone and periodontium differentiating toward osteoblasts, thus contributing to the protective actions during AP (*Figure 5H*).

We investigated the heterogeneity of MSCs by applying branch expression analysis modeling (BEAM) and corresponding pseudotime analysis implemented in Monocle 2 (*Qiu et al., 2017a*). The results revealed three distinct states. Of these, state 2 and state 3 represented two diverse differentiated cell populations (terminal branch; *Figure 6A*). We then examined the characteristics of the MSCs with respect to their specific states. We plotted the estimated pseudotime of each cluster cell in the state space, and the results indicate that starting with MSC_endothelial cells, MSCs were able to differentiate into MSC_OLCs and MSC_neurological cells. MSC_inflammatory cells may correspond to an intermediate state (*Figure 6B*). Interestingly, MSC_OLCs in state 3, as osteogenic primed MSCs,

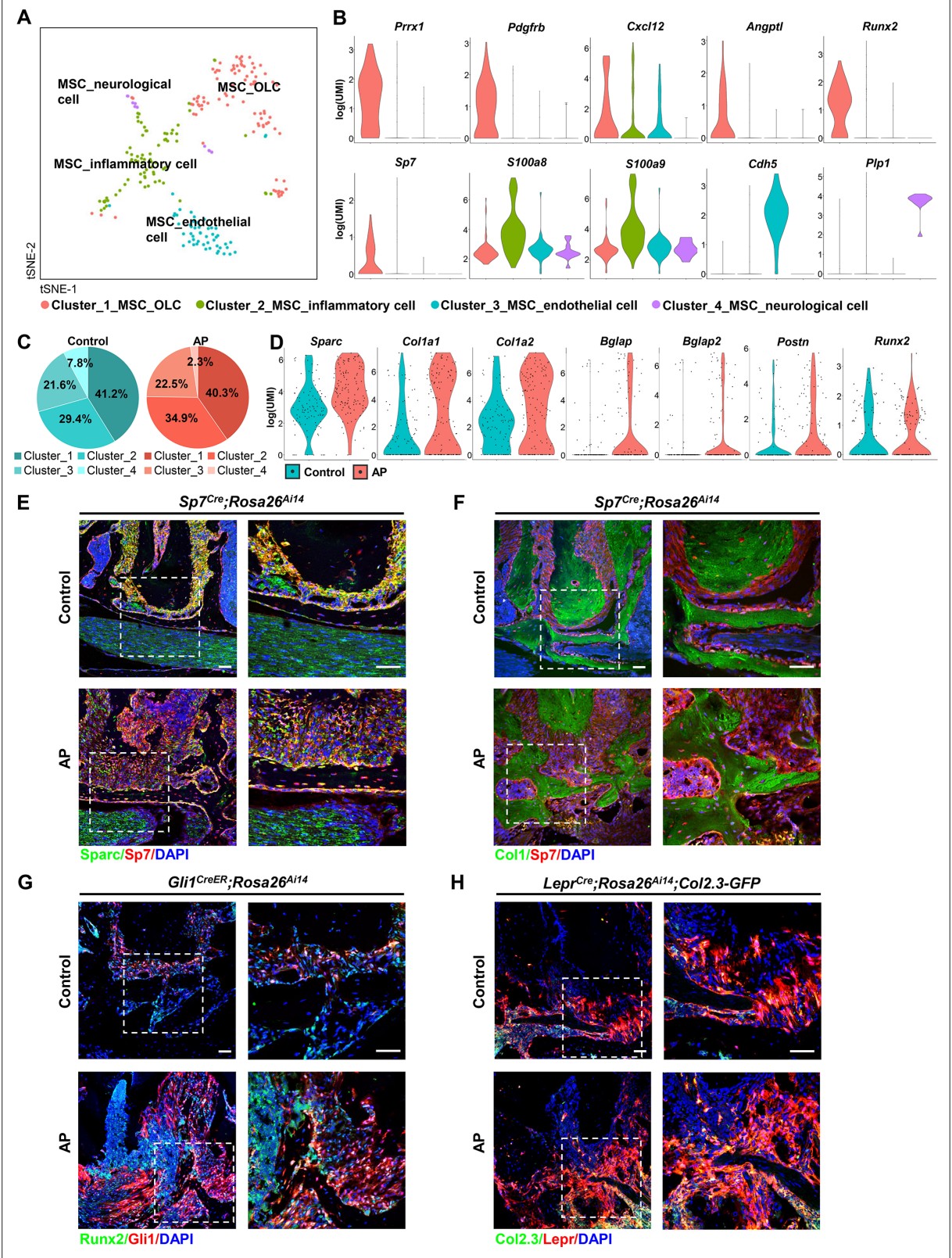

**Figure 5.** Identification and characterization of apical periodontitis (AP)-associated mesenchymal stem cell (MSC) population and its subclusters. (A) t-Stochastic neighbor embedding (t-SNE) representation an unsupervised clustering of single cells within the MSC cluster. (B) Violin plots of MSC subcluster-specific expression of representative genes. (C) The percentages of four subclusters of MSC population were quantified in control and AP groups. (D) Violin plots depict the changes in the expression of top upregulated genes in the MSC cluster. (E and F) Immunofluorescence double

*Figure 5 continued on next page*

*Figure 5 continued*

staining of Sparc (**E**) and Col1 (**F**) in Sp7-expressing osteoprogenitors. Scale bar, 50 µm. (**G**) Immunofluorescence double staining of Runx2 in Gli1⁺ periodontal stem cells (PDLSCs). Scale bar, 50 µm. (**H**) Lineage tracing analysis of Lepr⁺ PDLSCs and Col2.3⁺ osteoblasts. Scale bar, 50 µm.

The online version of this article includes the following figure supplement(s) for figure 5:

**Figure supplement 1.** Red fluorescent protein expression in *Prrx1^Cre^;Rosa26^Ai14^*, *Sp7^Cre^;Rosa26^Ai14^*, *Gli1^CreER^;Rosa26^Ai14^*, and *Lepr^Cre^;Rosa26^Ai14^;Col2.3-GFP* mouse models.

**Figure supplement 2.** Gene Ontology (GO) enrichment analysis of the biological functions of mesenchymal stem cell (MSC) subclusters.

**Figure supplement 3.** Real-time quantitative PCR (qPCR) and bulk RNA-seq analysis of osteogenic markers during apical periodontitis (AP) situation.

exhibited the highest tendency toward osteoblastic differentiation with upregulated expressions of *Col1a1*, *Col1a2*, *Bglap*, *Spp1*, and *Postn* (*Figure 6C*). Notably, this MSC population increased significantly in the AP group (*Figure 6D*). A comparison of the dynamics of the gene expression pattern across pseudotime revealed upregulation of osteogenic genes such as *Sparc*, *Col1a1*, *Col1a2*, and *Bglap* during inflammation when compared to the homeostatic state (*Figure 6E*). Thus, the MSC

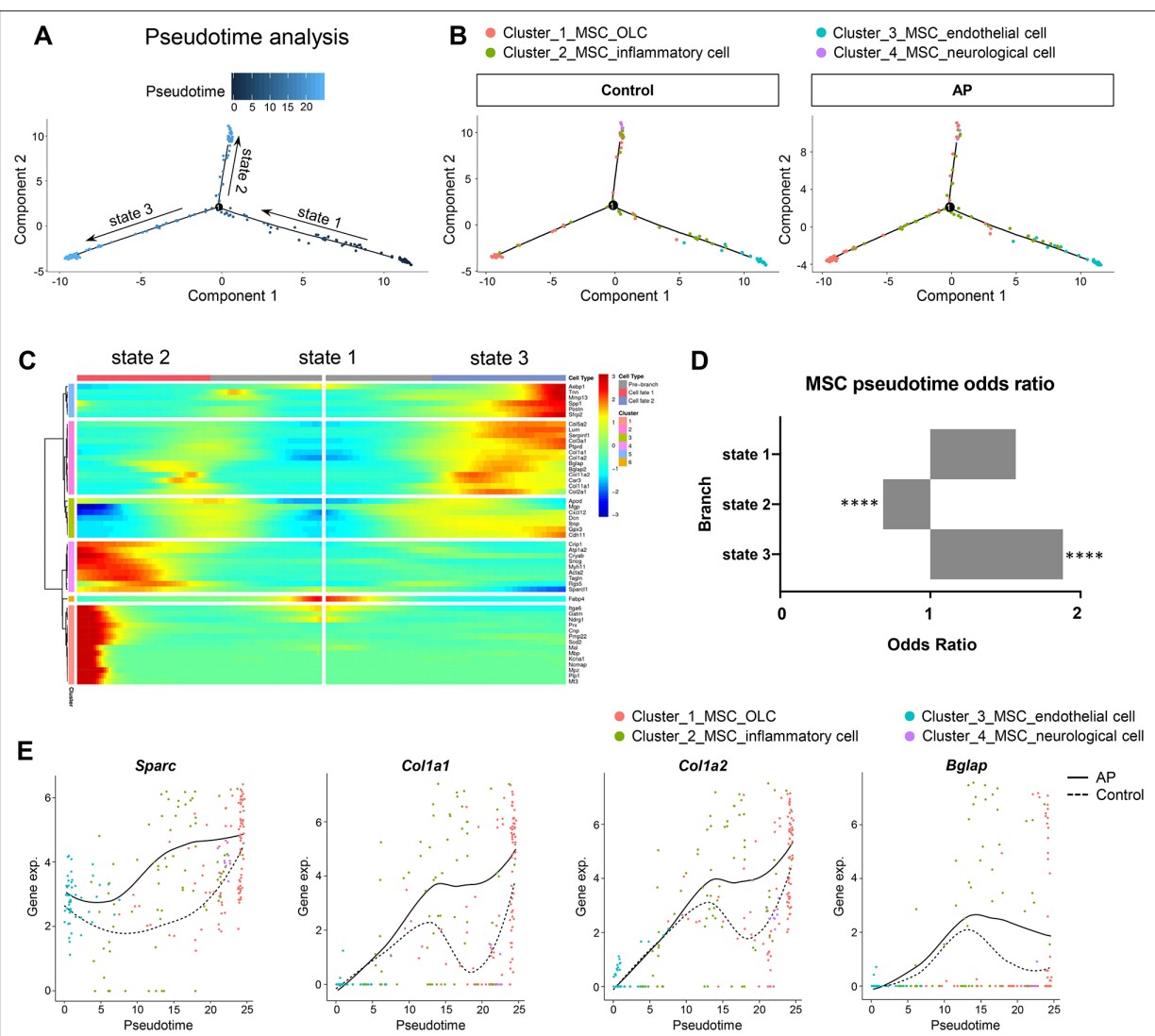

**Figure 6.** Apical periodontitis (AP) stimulates mesenchymal stem cell (MSC) differentiation toward osteoblast lineage cells. (**A**) Pseudotime analysis of the MSC population. (**B**) Reconstructed cell differentiation trajectory of MSC populations in control and AP groups. (**C**) Heatmap of differential genes of three states. (**D**) Bar plot of numerical changes between AP and control in respective states as identified in pseudotime analysis. Fisher's exact test with Bonferroni correction was used. ****p<0.0001. All data were exhibited as mean ± SEM. (**E**) Comparison between the pseudotime gene trajectories of MSC subpopulations showed upregulation of *Sparc*, *Col1a1*, *Col1a2*, and *Bglap*. Black line indicates AP, and dotted lines indicate control.

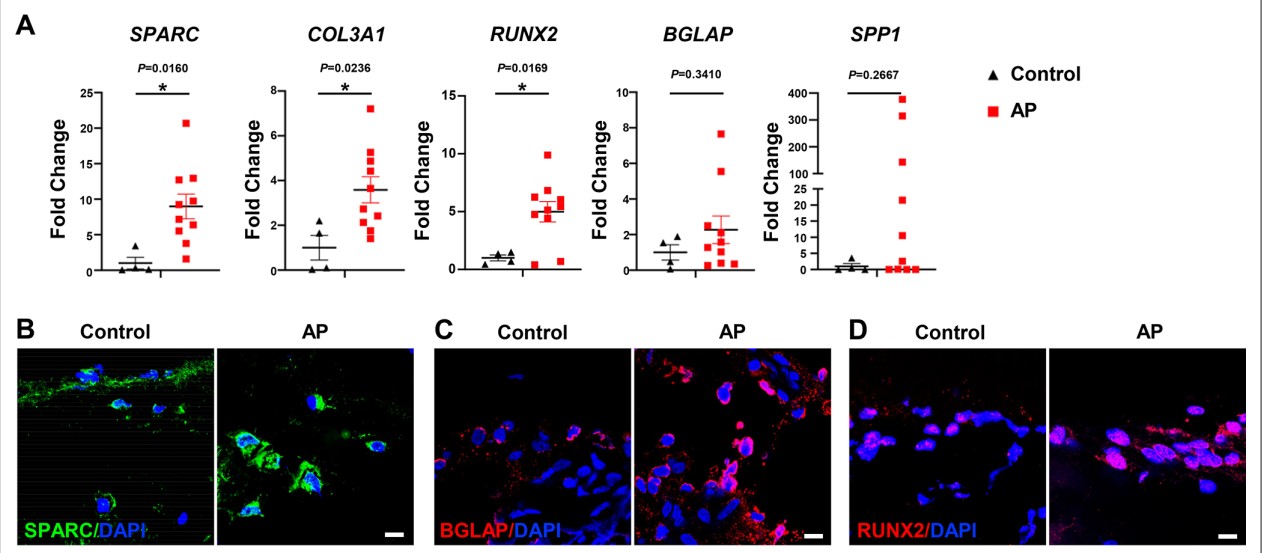

**Figure 7.** Human alveolar bone in apical periodontitis (AP) is associated with higher osteolineage subcluster (OLC)-associated transcripts. (**A**) Gene expression of *SPARC, COL3A1, RUNX2, SPP1*, and *BGLAP* in samples of healthy individuals and AP patients showed an upregulated expression in AP group. n=4 in healthy individuals and n=10 in AP patients. *p<0.05. All data are shown as the mean ± SEM. (**B–C**) Immunofluorescence staining of SPARC, BGLAP, and RUNX2 in bone marrow of human alveolar bone exhibited high osteogenic protein levels in AP. n=4. Scale bar, 10 μm.

subcluster exhibited gene expression levels of osteogenic markers that increased during AP. These results were in accordance with our previous single-cell analysis of gene expression in the AP group (*Figure 5D–H*).

## Higher OLC-associated marker levels were identified in alveolar bone from patients with AP

We next determined whether osteogenesis within the MSC population was altered in human patients with AP. The expressions of *SPARC, COL3A1*, and *RUNX2* were significantly enhanced in alveolar bone from patients with AP, accompanied by increases in *SPP1* and *BGLAP* (*Figure 7A*). Moreover, immunostaining detected a more extensive distribution of SPARC, BGLAP, and RUNX2 in AP alveolar bone marrow, confirming higher osteogenic potential under AP conditions (*Figure 7B–D*). Upregulation of OLC-associated markers in AP lesions from patients is consistent with our previous results, verifying an augmentation of osteogenic characteristics within the MSC subcluster in alveolar bone marrow under AP conditions.

In summary, the current study elucidated the characterization of MSCs and immunoresponsive cells under healthy and chronic AP conditions, including unforeseen heterogeneity in their states of activation. We believe that this analysis presents a comprehensive transcriptomic view of the alveolar bone microenvironment during AP. These results could pave the way for designing new therapeutic approaches by targeting MSCs to restore the alveolar bone lesions caused by AP and other inflammatory-related bone diseases.

## Discussion

In this study, we employed scRNA-seq technology for an unbiased assessment of cell populations in alveolar bone marrow from mice with induced chronic AP. We identified 15 discrete cell clusters, each with unique transcriptional signatures and putative functions during AP progression. Immune cells accounted for the majority of the cell components in the mandibular microenvironment. Cell-to-cell communication analysis revealed that among the multiple cell types, the MSC population had the most interaction with itself and other immune cells under inflammatory conditions. Furthermore, we identified four main subclusters within the MSC population. The transcriptional profiles of each subcluster from both the homeostatic and disease states were significantly different. We used the expression profile and the pseudotime analysis to demonstrate that MSCs were stimulated by inflammation and

differentiated toward osteogenic lineage cells in order to protect the alveolar bone. We confirmed the enriched genes and proteins associated with osteogenesis in AP-associated alveolar bone from both the murine models and human patients. Our results define the bone marrow microenvironment in a homeostatic state and under chronic AP, highlighting the protective action of MSCs in regeneration of alveolar bone lesions.

The oral cavity is one of the most complex microenvironments where host and microbes interacts closely (*Hasturk et al., 2012*). Oral inflammatory diseases such as AP is a complex disease characterized by the simultaneous presence of inflammation, destructive immunoreactions, and healing processes (*Márton and Kiss, 2000*; *Márton and Kiss, 2014*). A dense infiltration of immunocompetent cells is present in periradicular lesions. Extensive studies have examined these cells by immunostaining and gene expression techniques to determine their specific properties in terms of numbers, morphology, and function (*Graunaite et al., 2012*). These results provide a deeper understanding of immune cells and their inflammatory mediators. Yet the extent of their heterogeneity and the distinct markers of cell subsets have remained unexplored.

Our scRNA-seq analysis supplements knowledge of the landscape of bone marrow immunoreactions during oral inflammatory diseases. For example, the pro-inflammatory cytokines (IL-1β, IL-6, and TNF-α) have been recognized as key mediators associated with the persistence of the disease (*Morsani et al., 2011*). We found high expression of *Il1b* and *Il6* in the mast cell cluster and *Tnf* in the monocyte population. The release of these substances is directly related to both inflammatory events and bone resorption (*Bracks et al., 2014*). Moreover, other inflammatory cells, including neutrophils, macrophages, and lymphocytes, present specific expression patterns of various cytokines and chemokines. In particular, a significant upregulation of *Cxcl2* expression was identified in neutrophil and macrophage populations. Cxcl2 was detected in endodontic infections (*de Gomes and Herrera, 2018*) and is one of the most critical chemokines for neutrophil recruitment at a site of inflammation (*Kobayashi, 2008*; *Silva et al., 2007*). Increased levels of Cxcl2 have been reported to be associated with bone-resorptive activity in periapical lesions (*AlShwaimi et al., 2013*), highlighting the pro-inflammatory role of Cxcl2 in the progression of AP. Furthermore, we detected several new anti-inflammatory factors such as *Ifitm1* and *Ifitm2* in monocyte/macrophage subsets. Ifitm1 and Ifitm2 encode for a family of small transmembrane proteins localized in the plasma and endolysosomal membranes. Although the function of Ifitm has not been assessed in the context of AP, there is considerable evidence suggesting that Ifitm1 and Ifitm2 can be stimulated by interferons and exert an anti-inflammatory function in innate and adaptive immunity (*Liao et al., 2019*; *Yánez et al., 2020*). It is notable that the frequencies of macrophage, T cell, B cell, NK cell, epithelial cell, and MSCs were significantly increased in AP samples. Being one of the most significant components of the innate immune system, macrophages function to remove the pathogenic substances and dead cells by phagocytosis (*Lu et al., 2021*). They can be divided into two subsets, M1-like macrophages and M2-like macrophages. The M1 type is termed as the proinflammatory one, which is Cd86, Cd80, Il1b, and Tnf positive. M1-like macrophages also work as antigen presenting cells to interact with T helper 1 (Th1) cells (*Song et al., 2022*). Moreover, they release TNF-α, IL-1α, IL-6, and prostaglandins to increase the resorptive intensity in AP (*Paula-Silva et al., 2020*). By contrast, M2-like macrophages display immunoregulatory functions. They are positive for Cd163, Cd206, Msr1, and Il10. The analysis of macrophage subsets showed the M1-like macrophage accounted for the vast majority in AP lesions. The expression pattern of M2 markers were also presented in macrophage cluster (*Figure 3—figure supplement 1A, B*). Lymphocytes including T cells and B cells are crucial in the adaptive immune response of AP (*Wang et al., 2022*). Th1 cells are believed to be pro-inflammatory, while Th2 cells usually function as the immunosuppressive subcluster. Tregs are another important type of T cells related to the regenerative process of AP. They can release IL-10 and tissue growth factor (TGF)-β to ameliorate the local inflammation. Tregs also function as the suppressor of T cell response and restrict the progression of AP. Different types of B cells act conversely during AP process. It was reported that antigen-activated B cells and switched memory B cells increased RANKL expression then stimulated osteoclast generation and alveolar bone resorption (*Han et al., 2018*; *Settem et al., 2021*). In contrast, the plasma cells play a fundamental role in tissue repair by producing antibodies (*Han et al., 2018*; *Polanco et al., 2021*). NK cells exert a crucial function in host defense by rapidly producing cytokines and upregulating the expression of cell death-inducing molecules to eliminate infected cells (*Kratzmeier et al., 2023*). CD57[+] NK cells were presented in chronic AP lesions, and upregulated NK

cells were observed in periapical granulomas compared with radicular cysts (*Silva et al., 2017*). Proliferating epithelium is frequently present in inflamed apical lesions, which is usually developed from epithelial rests of Malassez (*Langeland et al., 1977*; *Lin et al., 2007*). It is an important protective bioactivity since these epithelial tissues aim to build a barrier to prevent the diffusion of microorganisms. Increased number of epithelial cells is highly connected with the progression toward periapical cyst, the connective tissue capsule of which is lined by a non-keratinized stratified squamous epithelium (*Schulz et al., 2009*). Our scRNA-seq analysis has revealed novel transcriptional signatures of specific cell populations which were not visible with bulk analysis. Further studies will be required to investigate the contribution of these factors to the immune response and inflammation.

Cell-cell communications mediated by ligand-receptor complexes are crucial to control diverse biological processes, including development, differentiation, and inflammation (*Efremova et al., 2020*). We employed CellPhoneDB to demonstrate previously unrecognized intercellular communication among all cell subsets, including close crosstalk among MSCs, macrophages, and dendritic cells. More importantly, CrossTalkeR enabled us to identify cell communication differences between two biological states: disease versus homeostasis (*Nagai et al., 2021*). Our results suggest that most ligand-receptor interactions were markedly reduced during AP. Alterations in cell-cell communication have also been described for multiple tissues and organs under other disease, injury, and infection conditions (*Armingol et al., 2021*). The shifts in the pattern of CCIs in the alveolar bone microenvironment may imply reorganization of the network in response to inflammation. It is worth noting that MSCs possessed the highest number of interactions with themselves and other cell types under both homeostasis and AP conditions, indicating MSCs act as a hub in CCIs. Thus, alterations of MSCs may substantially affect the biological behaviors of other cell types in both healthy and diseased states, further underscoring the importance of MSCs in alveolar bone marrow during inflammation.

In most cases of chronic AP, periapical bone destruction is self-limiting in which a new equilibrium has been established between root canal pathogens and anti-infective defense mechanisms (*Wang et al., 2011*). The mechanisms that underly the protective actions of immune cells in apical inflammatory tissues have been extensively studied (*Márton and Kiss, 2000*; *Márton and Kiss, 2014*; *Nair, 2004*; *Stashenko, 1990*). Yet, the kinetics for the healing of these lesions by regenerating alveolar bone is less well understood.

MSCs are recognized as a promising resource for periapical bone healing in the treatment of oral inflammatory diseases. Several lines of evidence support this tenet. The evoked-bleeding step in revascularization procedures for regenerative endodontic treatment leads to a substantial influx of MSCs, confirming the existence and regenerative properties of MSCs in the periapical region. Indeed, several dental MSCs are present in the periapical tissues, including MSCs derived from alveolar bone, periodontal ligament, and apical papilla (*Zhao and Chai, 2015*). More importantly, MSCs were found to be present in inflamed periapical tissues in adult patients with AP. Evidence showed that human apical papilla and its resident cells from teeth with pulp necrosis and AP could survive and retain their stemness in an inflammatory environment (*Chrepa et al., 2017*). MSCs found in the periapical region were positive for typical stem cell markers, such as STRO-1, CD90, CD73, CD105, CD146, and SOX2 (*Chrepa et al., 2015*; *Estrela et al., 2017*; *Liao et al., 2011*). These progenitors were able to form CFU-Fs, illustrating a typical MSC feature. However, our understanding of the characteristics of MSCs in inflamed periapical tissues remains at an early stage. In this study, we explored the heterogeneity and expression signature of MSCs in the presence of inflammation. Several key observations were only possible by study at the single-cell level. First, we identified four main subclusters in the MSC population: MSC_OLCs, MSC_endothelial cells, MSC_neurological cells, and MSC_inflammatory cells. The first three clusters were consistent with previously described MSC populations in alveolar bone marrow (*Lin et al., 2021*). These MSC subclusters have distinct differentiation potentials and may be associated with the pathogenesis of AP. Importantly, we noted that the proportion of MSC_inflammatory cells in AP was higher and accompanied by significantly increased expression of *S100a8/a9*. These factors are involved with $Ca^{2+}$ binding which is significantly increased during inflammatory processes (*Wang et al., 2018*). Recent studies have found that although MSCs do not express S100a8/a9 under healthy conditions, they do upregulate its expression in disease states (*Leimkühler et al., 2021*). The higher S100a8/a9 expression in the MSC_inflammatory cell subcluster suggests that S100a8/a9 might be a candidate for detecting an inflammatory-associated MSC state in AP. Activated by inflammatory bone destruction, these MSCs with increased osteogenic potentials may rescue the bone

resorption process, which reach the equilibrium between bone formation and resorption then drive the progression of AP into stable states (*Márton and Kiss, 2014*). Since the pathological stimuli exists constantly, the protective actions can alleviate the bone loss to some extent. In clinical practice, root canal therapy (RCT) aims to disinfect and remove the pathogenic factors, which make the protective activities overweigh the destructive ones (*Lin et al., 2009*). The bone lesions of AP patients receiving RCT usually fully recovered with resolution of radiolucency after the inflammation is controlled in apical area (*Soares et al., 2006*). The healing of AP lesion is highly correlated with the osteogenic potential of inflamed MSCs (*Lin et al., 2009*). The differentiative traits of these cells and their role in inflamed periapical tissue remain to be explored. Deeper understanding of the features of MSCs in inflammatory microenvironment and the regulatory network underlying this situation will shed light on the MSC-based regenerative medicine for AP and other inflammatory bone diseases.

An important finding in the present study is the increased osteogenic potential in the MSC population during inflammatory-associated bone lesions. Several lines of evidence support this conclusion. The overall size of the MSC population was significantly increased and was accompanied by upregulated expression of key osteogenic-related genes. Moreover, pseudotime analysis detected a specific state progression toward osteogenesis in the MSC cluster. The numerical changes in this state were statistically higher in AP samples when compared to control. MSCs in this state harbored the highest osteoblastic differentiation potential, along with upregulated expressions of *Sparc*, *Col1a1*, *Col1a2*, *Runx2*, and *Bglap*. More importantly, we confirmed this observation in animal models and human patients with AP. By using a lineage tracing strategy, we identified increased percentages of Prrx1+, Gli1+, Lepr+ MSC, and Sp7+ progenitor cells with higher expression of Sparc, Col1a1, and Runx2 in the alveolar bone region surrounding periapical lesions. We show, for the first time, upregulated SPARC, RUNX2, COL3A1, and BGLAP in alveolar bone from human patients with AP. It is notable that Sparc was the most upregulated marker in the MSC population from AP samples. Sparc is one of the most abundant non-collagenous proteins expressed in bone. It has a critical role in osteoblast differentiation, procollagen processing, and assembly as well as mineralization (*Rosset and Bradshaw, 2016*). SPARC mutations have been identified in patients with osteogenesis imperfecta and idiopathic osteoporosis (*Delany et al., 2008*; *Mendoza-Londono et al., 2015*). Sparc null mice exhibit decreased osteoblast number and activity along with reduced bone formation rate (*Delany et al., 2003*). Recent research using scRNA-seq revealed high levels of SPARC expression in human periodontal MSCs, highlighting the potential role of SPARC in oral MSC populations (*Pagella et al., 2021*). Notably, Sparc is also a molecule closely related to inflammation. Overexpressed Sparc is observed in inflammatory pulp cells, experimental glomerulonephritis, and ovarian cancer associated inflammation (*Dimitrova-Nakov et al., 2014*; *Pichler et al., 1996*; *Said et al., 2008*). Both pro-inflammatory and anti-inflammatory properties of Sparc have been reported (*Said et al., 2008*; *Sangaletti et al., 2011*; *Tanaka et al., 2018*; *Toba et al., 2015*). Sparc may act in two ways by activating different pathways depending on context. It therefore creates a microenvironment suitable for tissue remodeling and repair at different stages (*Ghanemi et al., 2020*). Given the upregulation of Sparc observed in AP mouse models and human patients, we believe that the MSC population was stimulated by inflammation and underwent osteogenesis with a significant function in bone matrix assembly and mineralization, thus serving as an important protective component in healing the bone lesions in inflammation.

In summary, we profiled the transcriptome of alveolar bone marrow single cells from control and AP samples, providing novel insights into the inflammatory biomarkers associated with the pathogenesis of oral inflammatory diseases. The interplay among various cell populations highlights the role of MSC subsets and the therapeutic potential of MSCs in healing bone lesions. These results demonstrate marked heterogeneity in the alveolar bone marrow microenvironment and provide a basis for considering MSC-based treatment for inflammatory-related bone diseases.

## Materials and methods
### Animal experiments

*Gt(ROSA)26Sor*$^{tm14(CAG-tdTomato)Hze}$ mice (*Rosa26*$^{Ai14}$; Cat# JAX:007914), *Sp7*$^{Cre}$ (Cat# JAX:006361), *Prrx1*$^{Cre}$ (Cat# JAX:005584), and *Gli1*$^{CreER}$ (Cat# JAX:007913) mice were purchased from Jackson Laboratory. *Col2.3-GFP* and *Lepr*$^{Cre}$ mice were kindly provided by Dr. Bo O. Zhou (Institute of Biochemistry and Cell Biology). Hybrid mice were generated to mark *Sp7-*, *Prrx1-*, *Lepr-*, and *Gli1-*expressing cells by

crossing *Sp7^Cre^*, *Prrx1^Cre^*, *Lepr^Cre^*, and *Gli1^CreER^* with *Rosa26^Ai14^* respectively. *Lepr^Cre^;Rosa26^Ai14^;Col2.3-GFP* mice were generated by crossing *Lepr^Cre^;Rosa26^Ai14^* mice with *Col2.3-GFP mice. Gli1^CreER^;Rosa26^Ai14^* mice (at postnatal day 50) were injected intraperitoneally with tamoxifen (Sigma-Aldrich) at dosage of 2.5 mg/10 g body weight every 2 d for three times. Wild-type mice were purchased from Chengdu Dossy Biological Technology Co., Ltd. All animal experiments were carried out in accordance with the guidelines of the Institutional Animal Care and Use Committee at the State Key Laboratory of Oral Diseases, Sichuan University (WCHSIRB-D-2021–339).

## AP mouse model

The AP model was generated using 8-wk-old mice as described previously (*Taira et al., 2019*). In brief, the pulp chambers of the mandibular first molars were opened by a high-speed handpiece with #1/4 carbide round bur under direct visualization by a stereoscopic microscope (Leica). A #10 endodontic K file was then used to probe the root canals. The teeth were then exposed to the oral environment for 3 wk.

## Single-cell sample preparation and sequencing

20 C57/B6 male mice with AP or control mice were used to acquire single-cell suspensions. Mandibles were dissected under a stereo microscope (Leica). Specifically, soft tissues, molars, incisors, and bone from behind the condyle were removed. Then, mandibular alveolar bone tissue was cut into small pieces (<1 mm³) and digested with 3 mg/mL collagenase type I (Gibco) and 4 mg/mL dispase II (Sigma) for 60 min at 37°C as previously described (*Cui et al., 2020*; *Yamaza et al., 2011*). Red blood cells were lysed with 1 mL Red Cell Lysis Solution (Biosharp). Cells were centrifuged and resuspended in 1% BSA solution. The final concentration of cells was $1 \times 10^6$ cells/mL. Cellular suspensions were loaded into Chromium microfluidic chips and barcoded with a 10× Chromium Controller (10× Genomics). RNA from the barcoded cells was subsequently reverse-transcribed and sequencing libraries constructed with reagents from a Chromium Single Cell 3' v3 reagent kit (10× Genomics) and sequenced with the NovaSeq system (Illumina).

## Preprocessing of scRNA-seq data

We used Cellranger (v3.1, 10× Genomics) to compare reads to the mouse genome mm10 and for detection of cells using default parameters. Next, we used the Seurat package (v3.1) for further analysis of the scRNA-seq. A gene with less than 3 cells, mitochondrial genes higher than 20%, cells with less than 200 expressed genes, and cells with erythrocyte gene expression higher than 5% were filtered out. The Seurat package was used for data normalization and dimensional reduction. Principal component analysis was based on the highly variable genes, and subclusters of cells were identified using specific gene expression in each group.

## Pseudotime analysis

The cluster identified as MSC was loaded into R environment and then the R package Monocle 2 was used to infer the trajectory and reduce the dimensional space to discover the state transitions of MSCs (*Qiu et al., 2017b*). Each cell's pseudotime was automatically calculated, and the branch was assigned on the principal graph. BEAM was run to analyze the hub genes in branches, and top 50 genes were visualized with the plot_genes_branched_heatmap function.

## CCI analysis

CellPhoneDB v.2.0 was performed to predict enriched cellular interaction between different cell populations according to the expression of a receptor by one state and a ligand by another one (*Efremova et al., 2020*). Receptors and ligands were analyzed when they were expressed in at least 10% cells of the corresponding cluster in mouse data. Subsequently, CrossTalkeR was used to visualize the ligand-receptor networks, which identified relevant ligands, receptors, and cell types contributing to changes in cellular crosstalk when contrasting AP and homeostasis states (*Nagai et al., 2021*). We then ranked the individual ligand by their number of interactions and normalized to the number of all possible CCIs to identify the most influential ligand. Interactions associated with Fn1 and Sele were visualized via Sankey plot.

## Bulk RNA-seq analysis

Healthy and AP mandibles were used to extract total RNA and analyzed with bulk RNA-seq. Power-Lyzer 24 Homogenizer (Qiagen) was used for lysis and homogenization of alveolar bone samples, and Trizol (Invitrogen) was used for mRNA extraction according to the manufacturer's protocol. NanoDrop ND-1000 (Thermo fisher Scientific) was used to quantify RNA concentration. Sequencing libraries were generated using the NEBNext UltraTM RNA Library Prep Kit for Illumina (NEB, USA), and index codes were added to correlate sequences to each sample. The library preparations were sequenced on an Illumina Novaseq 6000 (LC-Bio Technology Co., Ltd.).

## Quantitative real-time PCR

PrimeScript RT reagent kit (Takara) was used to reverse transcribed RNA, and qRT-PCR was performed by SybrGreen Supermix (Bio-Rad Laboratories). Relative expression was calculated using a $2^{\Delta\Delta CT}$ method with *Actb* for normalization. All primers are listed in the ***Supplementary file 1***.

## Collection of human alveolar bone

This study was permitted by the Ethical Committees of the West China Hospital of Stomatology, Sichuan University (WCHSIRB-D-2021–292). Written informed consent was obtained from all patients. Patients who were characterized with periapical rarefaction in radiography, suffered with a failed root canal treatment, and needed endodontic surgery were recruited as subjects. Patients with periodontitis, vertical root fracture, suspected odontogenic tumor, systematic diseases, or a history of antibiotic, antiviral, or immunosuppressive therapy for 3 mo were excluded. 10 human alveolar bone samples were collected from discards during the apicoectomy surgery. Normal apical bone tissue was harvested from patients who required a bone repair procedure. There was no significant difference in age or gender between the control group and the patients with AP. The samples were rinsed by sterile PBS. Subsequently, part of the samples was quick-frozen with liquid nitrogen and stored at –80°C for mRNA extraction. Others were fixed by 4% paraformaldehyde (PFA) and decalcified by 20% EDTA for immunostaining.

## Immunostaining

Immunofluorescent staining was performed to assess the expression of proteins in the region of the AP lesion. Mandibles from *Sp7^Cre;Rosa26^Ai14*, *Prrx1^Cre;Rosa26^Ai14*, *Lepr^Cre;Rosa26^Ai14;Col2.3-GFP*, and *Gli1^CreER;Rosa26^Ai14* mice were dissected and fixed in 4% paraformaldehyde for 3 d. Subsequently, samples were decalcified in 20% EDTA (pH 7.5) and embedded in Tissue-Tek O.C.T Compound (Sakura) and cut into 8 µm sections using CryoStar NX50 (Thermo fisher Scientific). Slides were permeabilized with 0.3% Triton X-100 (Beyotime) for 15 min, blocked with 5% BSA for 1 hr, incubated with anti-Sparc (1:100, R&D, AF942) or anti-Collagen I (1:200, Abcam, ab21286) or anti-Bglap (1:200, Abcam, ab93876) or anti-Runx2 (1:200, Abcam, ab23981) overnight at 4°C, and then incubated with Alexa fluor 488 (1:1000, Invitrogen, A11070) for 1 hr at room temperature. 4',6-diamidino-2-phenylindole dihydrochloride (DAPI) (Vector) was used as a nuclei counterstain. An Olympus confocal microscope FV3000 (Olympus) was used to capture images.

## Statistical analysis

GraphPad Prism 9.0 (GraphPad Software Inc) was used for statistical analysis. Unpaired two-tailed Student's t test was used in two-group comparisons. All data were exhibited as mean ± SEM. p-Values<0.05 were considered statistically significant for all analyses.

## Acknowledgements

This work was supported by NSFC grants 81800928, 81901040, 82222015 and 82171001, the Young Elite Scientist Sponsorship Program by CAST (No. 2020 QNRC001 and 2018QNR001), the Sichuan Science and Technology Program (No. 2019YJ0054), Research Funding from West China School/Hospital of Stomatology Sichuan University (No. RCDWJS2021-1), State Key Laboratory of Oral Diseases Open Funding Grant SKLOD-R013.

## Additional information

### Funding

| Funder | Grant reference number | Author |
|---|---|---|
| National Natural Science Foundation of China | 81800928 | Yi Fan |
| National Natural Science Foundation of China | 81901040 | Chenchen Zhou |
| National Natural Science Foundation of China | 82171001 | Chenchen Zhou |
| Young Elite Scientist Sponsorship Program by CAST | 2020QNRC001 | Chenchen Zhou |
| Young Elite Scientist Sponsorship Program by CAST | 2018QNR001 | Yi Fan |
| Sichuan Science and Technology Program | 2019YJ0054 | Yi Fan |
| Research Funding from West China School/ Hospital of Stomatology Sichuan University | RCDWJS2021-1 | Yi Fan |
| State Key Laboratory of Oral Diseases Open Funding Grant Sichuan University | SKLOD-R013 | Yi Fan |
| National Natural Science Foundation of China | 82222015 | Chenchen Zhou |

The funders had no role in study design, data collection and interpretation, or the decision to submit the work for publication.

### Author contributions

Yi Fan, Conceptualization, Data curation, Formal analysis, Supervision, Funding acquisition, Validation, Investigation, Methodology, Writing - original draft, Writing - review and editing; Ping Lyu, Formal analysis, Validation, Investigation, Methodology, Writing - original draft; Ruiye Bi, Data curation, Formal analysis, Validation, Investigation, Methodology; Chen Cui, Data curation, Investigation, Methodology, Writing - original draft; Ruoshi Xu, Resources, Validation; Clifford J Rosen, Supervision, Validation, Writing - original draft, Writing - review and editing; Quan Yuan, Conceptualization, Resources, Data curation, Supervision, Funding acquisition, Writing - original draft, Project administration, Writing - review and editing; Chenchen Zhou, Conceptualization, Data curation, Formal analysis, Supervision, Funding acquisition, Validation, Investigation, Methodology, Writing - original draft, Project administration, Writing - review and editing

### Author ORCIDs
Yi Fan http://orcid.org/0000-0001-7511-9872
Ping Lyu http://orcid.org/0000-0001-9908-9989
Clifford J Rosen http://orcid.org/0000-0003-3436-8199
Chenchen Zhou http://orcid.org/0000-0002-6427-5869

### Ethics

Human subjects: This study was permitted by the Ethical Committees of the West China Hospital of Stomatology, Sichuan University (Permit Number: WCHSIRB-D-2021-292). Written informed consent was obtained from all patients.

All animal experiments were carried out in accordance with the guidelines of the Institutional Animal Care and Use Committee at the State Key Laboratory of Oral Diseases, Sichuan University (Permit Number: WCHSIRB-D-2021-339).

Decision letter and Author response
Decision letter https://doi.org/10.7554/eLife.82537.sa1
Author response https://doi.org/10.7554/eLife.82537.sa2

---

## Additional files

### Supplementary files

- Supplementary file 1. Quantitative RT-PCR (qRT-PCR) primer sequences.

- MDAR checklist

- Source data 1. Source data files for *Figure 1D*, *Figure 1—figure supplement 1B*, *Figure 2A*, *Figure 3I*, *Figure 4D-E*, *Figure 5C*, *Figure 5—figure supplement 3A-B*, *Figure 6D* and *Figure 7A*.

### Data availability

All data generated or analysed during this study are included in the manuscript and supporting file; Source Data files have been provided for Figures 1, 2, 3, 4, 5, 6, 7, Figure1-figure supplement 1 and Figure5-figure supplement 3. Sequence data are deposited in the NCBI Gene Expression Omnibus under accession numbers GSE212975 and GSE221990.

The following datasets were generated:

| Author(s) | Year | Dataset title | Dataset URL | Database and Identifier |
|---|---|---|---|---|
| Fan Y, Lyu P, Yuan Q, Zhou C | 2022 | Single-cell gene expression of control mandibular bone marrow cells and mandibular bone marrow cells under apical periodontitis | https://www.ncbi.nlm.nih.gov/geo/query/acc.cgi?acc=GSE212975 | NCBI Gene Expression Omnibus, GSE212975 |
| Fan Y, Lyu P, Yuan Q, Zhou C | 2023 | RNA-seq analysis of RNA extracted from control alveolar bone samples and alveolar bone under apical periodontitis | https://www.ncbi.nlm.nih.gov/geo/query/acc.cgi?acc=GSE221990 | NCBI Gene Expression Omnibus, GSE221990 |

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
