## [Editor Report]

Data from scRNA-Seq analysis demonstrated that acute inflammation stimulates periodontal stem cells to differentiate into osteoblast lineage cells to protect the alveolar bone. In murine models and patients with apical periodontitis, the genes and proteins associated with osteogenesis were enriched. The studies help us understand how MSCs respond to inflammation during apical periodontitis disease progression.

---

## [Decision Letter]

**Decision letter after peer review:**

Thank you for submitting your article "Creating an atlas of the bone microenvironment during oral inflammatory-related bone disease using single-cell profiling" for consideration by *eLife*. Your article has been reviewed by 3 peer reviewers, including Di Chen as the Reviewing Editor and Reviewer #1, and the evaluation has been overseen by Carlos Isales as the Senior Editor.

Essential revisions:

1) Trajectory analysis alone is not sufficient. Certain genetic mouse models (Gli1-CreERT2, Prrx1-CreERT2, LepR-Cre mice) may be used to identify periodontal stem cells.

2) The authors need to check Runx2 expression in MSCs during inflammation in the alveolar bone.

3) The authors need to carefully discuss and address other major comments raised by reviewers.

*Reviewer #1 (Recommendations for the authors):*

This is a well-conducted research project with comprehensive data, which provides full pictures of cell clusters and populations in alveolar bone during inflammation in AP mice or in patients with AP.

1) In MSCs of the mice with AP, the osteogenic genes, such as Col1a1, Col1a2, Bglap, Bglap2, and Postn, were upregulated. Since these genes are regulated by Runx2, did the authors check the changes in the expression of Runx2 in MSCs?

2) Similarly, in the alveolar bone of the patients with AP, the osteogenic marker genes, such as SPARC, Col3a1, and OCN, were also upregulated. These genes are also regulated by Runx2, authors need to check the changes in the expression of Runx2 in human alveolar bone too.

3) The frequencies of T cell, B cell, NK cell, macrophage, epithelial cell, and MSC were significantly increased in AP samples. The authors may discuss which type of cells plays a major role in the inflammation process of AP.

4) Please change Prx1-Cre into Prrx1-Cre in the text and in the figures.

5) Authors need to describe statistical methods used in figure legends.

*Reviewer #2 (Recommendations for the authors):*

The study is appropriate as a resource paper to be published on *eLife*.

The following issues should be addressed:

1. A major conclusion from the study based on scSeq analysis is that acute inflammation stimulates periodontal stem cells to differentiate towards osteoblast lineage cells. Trajectory analysis alone is not sufficient. Multiple models (Gli1-CreERT2, Prx1-CreERT2, LepR-Cre etc.) are currently available for identifying periodontal stem cells. I highly recommend the authors verify the above conclusion using one of these transgenic models.

2. In addition to the in vivo analysis, expression changes of SPARC, Col1a1, Col1a2 and Bglap in mice model proposed by gene trajectory analysis (Figure 5) should also be verified with either real-time PCR or western blot.

3. Despite of Prx1, multiple markers, including Gli1, LepR, aSMA etc., have been identified as the MSC markers for periodontal tissue. Were their corresponding populations within the AP periodontal tissue also changed?

---

## [Author Response]

Essential revisions:1) Trajectory analysis alone is not sufficient. Certain genetic mouse models (Gli1-CreERT2, Prrx1-CreERT2, LepR-Cre mice) may be used to identify periodontal stem cells.2) The authors need to check Runx2 expression in MSCs during inflammation in the alveolar bone.3) The authors need to carefully discuss and address other major comments raised by reviewers.

We thank the editor for these valuable comments. We added the lineage tracing analysis of periodontal stem cells using *Prrx1Cre;Tm^fl/+^, LepRCre;Tm^fl/+^;Col2.3-GFP* and *Gli1creER;Tm^fl/+^* mice. We performed immunofluorescent double staining to study the differential fate of periodontal stem cells. Both human samples and mouse models have been used to investigate the expression of Runx2 during AP. The results of RNA-seq analysis, real-time qPCR and immunofluorescent staining help to verify that Runx2 was involved in the regulatory network with increased expression during AP. Other major comments raised by reviewers have also been discussed and addressed with experiments.

Reviewer #1 (Recommendations for the authors):This is a well-conducted research project with comprehensive data, which provides full pictures of cell clusters and populations in alveolar bone during inflammation in AP mice or in patients with AP.1) In MSCs of the mice with AP, the osteogenic genes, such as Col1a1, Col1a2, Bglap, Bglap2, and Postn, were upregulated. Since these genes are regulated by Runx2, did the authors check the changes in the expression of Runx2 in MSCs?

Thank you for this valuable suggestion to focus on the role of Runx2. We observed a trend towards increased Runx2 expression in MSC cluster by using Single-cell RNA-seq study (Figure 5D). We also performed bulk RNA-seq using mandibular bone samples from sham and AP mice. The results suggested a significantly upregulated Runx2 expression in AP mandibles (Figure 5—figure supplement 3B). Real-time quantitative PCR and immunofluorescent staining further confirmed the higher Runx2 expression under AP situation (Figure 5—figure supplement 3A). Moreover, increased number of Runx2^+^-osteoblasts were observed upon AP stimulation in vivo (Figure 5G).

2) Similarly, in the alveolar bone of the patients with AP, the osteogenic marker genes, such as SPARC, Col3a1, and OCN, were also upregulated. These genes are also regulated by Runx2, authors need to check the changes in the expression of Runx2 in human alveolar bone too.

We analyzed the expression of Runx2 in human alveolar bone at transcript and protein levels. There was a significantly higher Runx2 expression in the alveolar bone of AP patients compared to healthy individuals, which was consistent with the results observed in mouse models, confirming an augmentation of osteogenic characteristics in alveolar bone marrow under AP conditions (Figure 7A, D).

3) The frequencies of T cell, B cell, NK cell, macrophage, epithelial cell, and MSC were significantly increased in AP samples. The authors may discuss which type of cells plays a major role in the inflammation process of AP.

We appreciate this point raised by the reviewer. T cell, B cell, NK cell and macrophage are crucial components of immune cells. Macrophages function to remove the pathogenic substances and dead cells by phagocytosis (Lu, Xu, Liu, and Zhang, 2021). We have analyzed and discussed two subsets of macrophages, the M1-like macrophages and M2-like macrophages. The results showed that the pro-inflammatory M1 macrophage accounted for the vast majority in AP lesions. With immunoregulatory function, M2 macrophages were also identified in the macrophage cluster (Figure 3—figure supplement 1A, B). The T lymphocyte family is composed of several subclusters (Wang et al., 2022). The Th1 cells are believed to be pro-inflammatory while Th2 cells usually function as the immunosuppressive subcluster. Tregs are another important type of T cells related to the regenerative process of AP. They can release IL-10 and tissue growth factor (TGF)-β to ameliorate the local inflammation. Tregs also function as the suppressor of T cell response and restrict the progression of AP. Different types of B cells act conversely during AP process. It was reported that antigen-activated B cells and switched memory B cells increased RANKL expression then stimulated osteoclast generation and alveolar bone resorption (Han, Jin, Miao, Shi, and Lin, 2018; Settem, Honma, Chinthamani, Kawai, and Sharma, 2021). In contrast, the plasma cells play a fundamental role in tissue repair by producing antibodies (Han et al., 2018; Polanco et al., 2021). NK cells exert a crucial function in host defense by rapidly producing cytokines and upregulating the expression of cell death-inducing molecules to eliminate infected cells (Kratzmeier, Singh, Asiedu, and Webb, 2022). CD57^+^ NK cells were presented in chronic AP lesions and upregulated NK cells were observed in periapical granulomas compared with radicular cysts (Silva et al., 2017). Furthermore, proliferating epithelium is frequently present in inflamed apical lesions, which is usually developed from epithelial rests of Malassez (Langeland, Block, and Grossman, 1977; L. M. Lin, Huang, and Rosenberg, 2007). These epithelial tissues exert important protective bioactivity since they aim to build a barrier to prevent the diffusion of microorganisms. Increased number of epithelial cells is highly connected with the progression towards periapical cyst, the connective tissue capsule of which is lined by a nonkeratinized stratified squamous epithelium (Schulz, von Arx, Altermatt, ad Bosshardt, 2009). The MSC population normally stay quiescent in adulthood until being activated by pathologic situations. In response to AP, increasing number of MSCs with higher osteogenic potential master the protective and regenerative process of the surrounding bone.

We added the related contents in the Discussion section.

4) Please change Prx1-Cre into Prrx1-Cre in the text and in the figures.

Thank you for your suggestion. We have revised the text and figures accordingly.

5) Authors need to describe statistical methods used in figure legends.

We have added the statistical methods used in figure legends.

Reviewer #2 (Recommendations for the authors):The study is appropriate as a resource paper to be published on eLife.The following issues should be addressed:1. A major conclusion from the study based on scSeq analysis is that acute inflammation stimulates periodontal stem cells to differentiate towards osteoblast lineage cells. Trajectory analysis alone is not sufficient. Multiple models (Gli1-CreERT2, Prx1-CreERT2, LepR-Cre etc.) are currently available for identifying periodontal stem cells. I highly recommend the authors verify the above conclusion using one of these transgenic models.

We appreciate this point raised by the reviewer. We generated AP mouse models using *Prrx1Cre;Tm^fl/+^*, *LepRCre;Tm^fl/+^;Col2.3-GFP* and *Gli1creER;Tm^fl/+^* mice to perform lineage tracing analysis. Results showed that increased numbers of Prrx1^+^ MSC_OLCs as well as upregulated Gli1^+^ and LepR^+^ periodontal stem cells (PDLSCs) in AP lesions (Figure 5—figure supplement 2A). Immunofluorescent staining further confirmed that Gli1^+^ PDLSCs migrated to AP lesions and differentiated into Runx2^+^-osteoblasts (Figure 5G), Moreover, using *LepRCre;Tm^fl/+^;Col2.3-GFP* mouse model, we identified more Tomato/GFP^+^ cells in the alveolar bone, confirming that inflammation could stimulate the differentiation of progenitor cells in the adjacent bone and periodontium towards osteoblasts, thus contributing to the protective actions during AP (Figure 5H). Considering Prrx1 is a transcription factor that is highly expressed during developmental stage of limb bud and craniofacial bone, previous lineage tracing study using *Prrx1-CreERT2* found the postnatal Prrx1^+^ cells were restricted in the continuously regenerating tissues, such as the incisor. As the expression of Prrx1 in the postnatal molar area is barely undetectable (Bassir et al., 2019; Martin and Olson, 2000; Wilk et al., 2017), we used *Prrx1Cre* to observe the Prrx1^+^ lineage cells in AP mouse models (Figure 5—figure supplement 1A).

Please see the revised data in figure 5, figure 5—figure supplement 1 and 2.

2. In addition to the in vivo analysis, expression changes of SPARC, Col1a1, Col1a2 and Bglap in mice model proposed by gene trajectory analysis (Figure 5) should also be verified with either real-time PCR or western blot.

Thank you for your valuable point. We verified their expressions by real-time qPCR and bulk RNA sequence analysis. The results revealed that Sparc, Col1α1, Col1α2 was significantly upregulated and Bglap tended to increase during AP (Figure 5—figure supplement 3A, B).

3. Despite of Prx1, multiple markers, including Gli1, LepR, aSMA etc., have been identified as the MSC markers for periodontal tissue. Were their corresponding populations within the AP periodontal tissue also changed?

We thank the reviewer for this comment and agree that PDLSCs also exert crucial functions during AP. In order to focus on the bone marrow microenvironment, we have extracted the teeth before subjected to scRNA-seq analysis as previously described (W. Lin et al., 2021). Therefore, the expression levels of PDLSCs markers such as Gli1, LepR, aSMA were relatively low in MSC cluster. Please see Author response image 1. Yet, we have utilized *LepRCre*, *Gli1creER* mice to perform lineage tracing analysis to investigate the role of PDLSCs during AP in vivo. Please refer the results to the response in comment 1.

**Author response image 1. sa2fig1:**